# Rethinking Adversarial Training with A Simple Baseline

## Abstract

We report competitive results on RobustBench for CIFAR and SVHN using a simple yet effective baseline approach. Our approach involves a training protocol that integrates rescaled square loss, cyclic learning rates, and erasing-based data augmentation. The outcomes we have achieved are comparable to those of the model trained with state-of-the-art techniques, which is currently the predominant choice for adversarial training. Our baseline, referred to as *SimpleAT*, yields three novel empirical insights. (i) By switching to square loss, the accuracy is comparable to that obtained by using both *de-facto* training protocol plus data augmentation. (ii) One cyclic learning rate is a good scheduler, which can effectively reduce the risk of robust overfitting. (iii) Employing rescaled square loss during model training can yield a favorable balance between adversarial and natural accuracy. In general, our experimental results show that SimpleAT effectively mitigates robust overfitting and consistently achieves the best performance at the end of training. For example, on CIFAR-10 with ResNet-18, SimpleAT achieves approximately 52% adversarial accuracy against the current strong AutoAttack. Furthermore, SimpleAT exhibits robust performance on various image corruptions, including those commonly found in CIFAR-10-C dataset. Finally, we assess the effectiveness of these insights through two techniques: bias-variance analysis and logit penalty methods. Our findings demonstrate that all of these simple techniques are capable of reducing the variance of model predictions, which is regarded as the primary contributor to robust overfitting. In addition, our analysis also uncovers connections with various advanced state-of-the-art methods.

## 1 Introduction

Recent research (Li et al., 2023) has shown that adversarial examples or out-of-distribution examples pose a serious threat to the robustness of deep models, leading to significant security issues. Consequently, there is an increasing demand for algorithms that can improve the robustness and generalization of these models. To this end, various methods have been proposed to defend against different types of adversarial attacks (Miller et al., 2020; Ortiz-Jiménez et al., 2021). Among them, adversarial training has been shown to be the most effective method by Athalye et al. (2018), who demonstrated its efficacy through empirical evidence. This approach involves training a robust model using min-max optimization (Madry et al., 2018).

In a nutshell, they can be categorized into various categories, including data augmentation (Alayrac et al., 2019; Carmon et al., 2019; Sehwag et al., 2021; Rebuffi et al., 2021a;b; Li & Spratling, 2023; Wang et al., 2023), architecture designing (Xie et al., 2019; Xiao et al., 2019; Chen et al., 2020a; Wu et al., 2021), input transformation (Yang et al., 2019; Dong et al., 2020; Mao et al., 2021; Yoon et al., 2021), certified defenses (Cohen et al., 2019; Salman et al., 2020), and adversarial training (Madry et al., 2018; Kannan et al., 2018; Zhang et al., 2019; Wang et al., 2019; Zhang et al., 2021a; Pang et al., 2022; de Jorge et al., 2022; Dong et al., 2022; Liu et al., 2023). While these approaches have achieved satisfactory results as evidenced by the leaderboard on RobustBench (Croce et al., 2021), the training objective has remained largely unchanged, with cross-entropy loss being the default choice. Recent studies have shown that using cross-entropy loss for a deep classifier is problematic (Guo et al., 2017; Pang et al., 2020a; Yu et al., 2020).

To this end, recent research (Hui & Belkin, 2021) has demonstrated through numerous experiments that deep neural networks trained with square loss can achieve comparable or even better performance than those

trained with cross-entropy loss. Furthermore, Hu et al. (2022) have recently analyzed the theoretical reasons behind why square loss works well, using the neural tangent kernel technique. They also revealed that the square loss provides benefits in terms of robustness and calibration. However, while improvements have been observed, they are small when compared to various advanced state-of-the-art methods for defending against adversarial attacks. That this raises a question: **Can square loss be effective for adversarial training as well?** On the other hand, recent methods usually follow a classical experimental setting as in Rice et al. (2020), where stochastic gradient descent with multiple step decays is used. Therefore, this raises another question: **Are there other better learning rate schedules?**

This paper introduces a simple baseline to answer these questions, called SimpleAT. It remarkably outperforms various cutting-edge methods that rely on cross-entropy loss. Specifically, SimpleAT highlights the significance of square loss, cyclic learning rates, and erasing-based data augmentations in enhancing model robustness and generalization. We conducted numerous experiments to investigate their impact and discovered several intriguing observations that offer novel perspectives for adversarial training. First, models trained with the square loss outperform those trained with cross-entropy loss and are equally competitive as models trained with cross-entropy loss together with erasing-based data augmentations (DeVries & Taylor, 2017; Zhong et al., 2020). Second, we find that combing cyclic learning rates with either erasing-based data augmentations or square loss can reduce robust overfitting while achieving comparable performance. This observation differs from previous works (Rice et al., 2020; Pang et al., 2021), which show that using each technique separately does not effectively mitigate the risk of robust overfitting. Third, models trained with the rescaled square loss always get better at both natural and adversarial accuracy, which is a sign of a good trade-off.

Generally speaking, no matter whether under PGD-based adversarial training (Madry et al., 2018) or FGSM-based adversarial training (de Jorge et al., 2022), our SimpleAT achieves its best performance at the end of training while achieving state-of-the-art results on CIFAR-10, CIFAR-100, and SVHN. For example, our SimpleAT can get about 52.3% adversarial accuracy and 85.3% natural accuracy on CIFAR-10 without using extra data, which demonstrates competitive results when compared to those reported in recent state-of-the-art methods (Rebuffi et al., 2021b; Li & Spratling, 2023). Moreover, we conduct experiments on corruption datasets such as CIFAR-10-C, where the model trained with SimpleAT can produce higher performance (with 2.7% accuracy gain) compared to that trained with the default settings in (Kireev et al., 2022).

Finally, we explain the reason why our proposed method is effective from two different theoretical perspectives: bias-variance theory and logit penalty methods. First, we utilize the adversarial bias-variance decomposition technique (Yu et al., 2021) to understand the generalization of our SimpleAT method. We observe a consistent phenomenon that the variance of the model trained using SimpleAT is extremely low, which indicates that SimpleAT can make adversarial training more stable. In other words, SimpleAT mainly reduces adversarial bias, which helps to improve the generalization. Second, we compare our SimpleAT with recent logit penalty methods to reveal that SimpleAT tends to reduce confidence in the true class. This is because overconfidence in adversarial examples can easily lead to robust overfitting (Liu et al., 2023) and worse generalization (Stutz et al., 2020).

This paper is organized as follows: Section 2 introduces the basic techniques we used that form the foundation of this work. Section 3 provides a comprehensive comparison and evaluation of related works alongside ours. Section 4 explains the reason why square loss makes sense from two theoretical perspectives. At last, Section **??** concludes this paper.

## 2 Methodology

### 2.1 Adversarial Training

The adversarial training (AT) paradigm underpins the majority of recently developed reliable mechanisms (Madry et al., 2018; Rice et al., 2020; de Jorge et al., 2022). Given a CNN model $\boldsymbol{f}_\theta(\cdot)$, an input image $\boldsymbol{x}$ with its ground-truth label $y$, AT is usually formulated as following *min-max optimization*:

$$\min_\theta \mathbf{L}_t(\boldsymbol{f}_\theta(\boldsymbol{x} + \boldsymbol{\delta}), y), \text{ s.t. } \boldsymbol{\delta} = \arg\max_{\boldsymbol{\delta}} \mathbf{L}_a(\boldsymbol{f}_\theta(\boldsymbol{x} + \boldsymbol{\delta}), y), \tag{1}$$

where the outer optimization is to train the parameter $\theta$ by minimizing the training loss $\mathbf{L}_t$, and the inner optimization is to generate the adversarial perturbation $\boldsymbol{\delta}$ by maximizing the attacked target function $\mathbf{L}_a$. Generally speaking, we typically use the widely-used cross-entropy loss function as both the training loss and attack target function, *i.e.*, $\mathbf{L}_t = \mathbf{L}_a$.

There are two main paradigms in AT: PGD-AT (Madry et al., 2018) and FGSM-AT (de Jorge et al., 2022). Both use stochastic gradient descent to update model parameters during the outer minimization optimization when input with adversarial examples. While the main distinction between them lies in the choice of the inner maximization method for generating the adversarial perturbation $\boldsymbol{\delta}$. Specially, PGD-AT uses the projected gradient descent-based adversarial attack (called PGD-attack (Madry et al., 2018)), which iteratively updates the perturbation from a random Gaussian noise $\boldsymbol{\delta}_0$. The formulation of PGD attack can be shown as:

$$\boldsymbol{\delta}_{i+1} = \boldsymbol{\delta}_i + \alpha \times \mathrm{sign}(\nabla \mathbf{L}_a(\boldsymbol{f}_\theta(\boldsymbol{x} + \boldsymbol{\delta}), y))), \tag{2}$$

$$\boldsymbol{\delta}_{i+1} = \max\{\min\{\boldsymbol{\delta}_{i+1}, \epsilon\}, -\epsilon\}, \tag{3}$$

where sign is the sign function, $\nabla \mathbf{L}_a$ is the gradient of target function *w.r.t.* input data, $\alpha$ is the step-size, and $\epsilon$ is the perturbation budget.

However, PGD-attack is not efficient for training because its cost increases linearly with the number of steps. To address this issue, a recent and promising solution is the FGSM-AT (Goodfellow et al., 2014; Wong et al., 2020). This approach approximates the inner maximization in Eq.(1) through single-step optimization, also known as FGSM-attack. Specially, the FGSM-attack calculates the adversarial perturbation along the direction of the sign of the gradient, using the general formulation:

$$\boldsymbol{\delta} = \eta + \alpha \cdot \mathrm{sign}(\nabla \mathbf{L}_a(\boldsymbol{f}_\theta(\boldsymbol{x} + \eta), y)), \tag{4}$$

$$\boldsymbol{\delta} = \max\{\min\{\boldsymbol{\delta}, \epsilon\}, -\epsilon\}, \tag{5}$$

where $\eta$ is drawn from a random uniform distribution within interval $[-\epsilon, \epsilon]$. However, most FGSM-AT methods suffer from catastrophic overfitting, in which a model becomes suddenly vulnerable to multi-step attacks (like PGD-attack). To this end, de Jorge et al. (2022) introduced a Noise-FGSM method, which learns a stronger noise around the clean sample without clipping process as follows:

$$\hat{\boldsymbol{x}} = \boldsymbol{x} + \eta, \tag{6}$$

$$\boldsymbol{x}_{adv} = \hat{\boldsymbol{x}} + \alpha \cdot \mathrm{sign}(\nabla \mathbf{L}_a(\boldsymbol{f}_\theta(\hat{\boldsymbol{x}}), y)), \tag{7}$$

where $\hat{\boldsymbol{x}}$ is the data augmented with additive noise, and $\boldsymbol{x}_{adv}$ is the generated adversarial example for outer minimization optimization. This new FGSM-AT method is further named NFGSM-AT.

## 2.2 Variations in Existing Training Protocols

Our goal is to study the essential components of existing training protocols, in comparison with previous methods. We identify three modifications to the training protocols that are essential, which include changes to the learning rate schedule, data augmentation, and loss function. These will be discussed in detail below.

**Learning rate schedule.** Most learning algorithms commonly use stochastic gradient descent (SGD) scheme to learn the weight parameter of neural network (such as parameter $\theta$ in Eq.1). Therefore, adjusting the learning rate, also called the learning rate schedule, is as important as selecting the actual algorithms.

Rice et al. (2020) conducted empirical studies on the impact of various learning rate schedules in enhancing robustness against adversarial examples. The schedules tested included piecewise decay, multiple decays, linear decay, cyclic scheduler, and cosine scheduler. Their findings revealed that all of these schedules are susceptible to robust overfitting. And they suggested that using a piecewise decay scheduler with an early-stopping technique consistently can achieve the best results. Moreover, Pang et al. (2021) verified these results that the piecewise decay scheduler is more suitable for PGD-AT.

Wong et al. (2020) found that one cyclic learning rate can work well under the FGSM-AT paradigm, where the learning rate linearly increases from zero to a maximum learning rate and back down to zero (Smith,

2017). This one cyclic learning rate is widely used in many following FGSM-AT schemes such as NFGSM-AT. That is, PGD-AT and FGSM-AT commonly adopt very different training protocols. On the other hand, some other works (Huang et al., 2020; Rebuffi et al., 2021b; Pang et al., 2022; Wang et al., 2023) use a cosine scheduler to achieve satisfactory results, but they work on learning a robust classifier by using semi-supervised learning or using extra dataset. This leads to the fact that it is difficult to determine that the actual performance improvement comes from a change in the learning rate because data augmentation or unlabeled datasets can also mitigate robust overfitting while achieving performance gain. But, generally speaking, we observe that recent defense methods (not only PGD-AT and FGSM-AT) commonly consider the cyclic learning rates[1]

**Data Augmentation.** Data augmentation is a technique used in machine learning to increase the size of a training set without requiring additional data samples. This can enhance the accuracy and robustness of models. However, it may not be effective in adversarial training as robust overfitting tends to occur, as noted by Rice et al. (2020). Although previous works (Rebuffi et al., 2021b; Li & Spratling, 2023; Liu et al., 2023) have used data augmentation to mitigate the risk of robust overfitting, in these methods, data regularization is often coupled with other factors and cannot clearly demonstrate its effectiveness. Interestingly, these methods generally adopt the same data regularization strategy, namely random erasing[2] (Zhong et al., 2020; DeVries & Taylor, 2017).

**Loss Function.** To optimize the objective in Eq.(1), it is important to select an appropriate loss function. While the cross-entropy loss is commonly used (both PGD-AT and FGSM-AT), recent research (Pang et al., 2022) has shown that a robust classifier trained with this loss may encounter issues such as robust overfitting and trade-offs. Therefore, careful consideration should be given when choosing the loss function.

On the contrary, a study by Hui & Belkin (2021) suggests that training with the square loss can produce comparable or even better results than cross-entropy in various machine learning tasks. Moreover, subsequent theoretical works explore the intriguing properties of models trained using square loss, such as robustness, generalization, neural collapse, and global landscape. These investigations employ innovative techniques like neural tangent kernel (Hu et al., 2022) or unconstrained feature models (Zhou et al., 2022).

However, these works mainly focus on standard image classification tasks where inputs are clean (natural) images. Although some works (Pang et al., 2022; Hu et al., 2022) have shown that using square loss helps achieve good robustness against adversarial examples, the performance gain is not satisfactory while there is still a significant gap to top-performing methods on RobustBench (Croce et al., 2021).

**Our Protocol.** Based on the analysis and observation of existing works, we define our **SimpleAT** protocol, which is independent of network architecture. Specially, we use SGD with momentum as the basic optimizer, and we set a momentum of 0.9 and a weight decay of $5 \times 10^{-4}$. We use the *one cycle learning rate*, and we set the maximum learning rate to 0.2. Unless otherwise stated, we set the number of the epochs for all methods as 200. We mainly use the IBDH data augmentation technique (Li & Spratling, 2023), which includes the *random erasing data augmentation*

Moreover, SimpleAT uses the Rescaled square loss function (shorten as *RSL*) (Hui & Belkin, 2021) that helps achieve superior performance, which is shown as follows:

$$L(\boldsymbol{x}, \boldsymbol{y}) = \|\boldsymbol{k} \cdot (\boldsymbol{f}(\boldsymbol{x}) - M \cdot \boldsymbol{y}_{hot})\|_2^2 = k \cdot (\boldsymbol{f}_y(\boldsymbol{x}) - M)^2 + \sum_{j=1, j \neq y}^{C} \boldsymbol{f}_j(\boldsymbol{x})^2. \tag{8}$$

We redefine the logit of a deep neural network as $\boldsymbol{f}$, which consists of $C$ components $\boldsymbol{f}_i$ (where $i = 1, 2, ..., C$) and $C$ represents the number of classes in a given dataset. The label vector is encoded using one-hot encoding and represented as $\boldsymbol{y}_{hot}$. The parameter $\boldsymbol{k}$ in Eq.(8) rescales the loss value at the true label while $M$ rescales the one-hot encoding.

**Note 1.** SimpleAT is a general adversarial training approach that uses a unified training protocol for PGD-AT and FGSM-AT. Our experiments show that SimpleAT can significantly improve performance while

---

[1]Here, we think the cosine scheduler also belongs to the cyclic learning rates.

[2]Random erasing (Zhong et al., 2020) and CutOut (DeVries & Taylor, 2017) are two almost identical data regularization techniques that are proposed during the same period. They differ slightly in their technical implementation, but their basic idea is the same, which is to remove some parts of an image. Therefore, this article refers to them collectively as random erasing.

maintaining a favorable equilibrium between robustness and accuracy, as well as reducing the risks of both robust and catastrophic overfitting.

**Note 2.** In our SimpleAT, the major difference between PGD-AT and FGSM-AT is the way that generates the adversarial examples in training. Therefore, for the PGD-AT, we use 10-step PGD-attack with $\alpha = 2/255$ to generate the adversarial examples, and the perturbation strength $\epsilon = 8/255$. For the FGSM-AT, we use NFGSM-attack to generate adversarial samples in training, and we follow the same experimental setting in (de Jorge et al., 2022) (where $\alpha = 8/255$ and $\epsilon = 8/255$).

**Note 3.** SimpleAT mainly employs two types of data augmentation techniques during the training phase. For CIFAR datasets, the first technique involves RandomCrop, RandomHorizontalFlip, and random erasing, while for SVHN, only RandomCrop and random erasing are used. We call this technique *default data augmentation* (DAA). The second one utilizes a robust data augmentation technique, called Improved Diversity and Balanced Hardness (IDBH), for all datasets. IDBH includes RandomHorizontalFlip, CropShift, ColorShape, and random erasing. All hyper-parameters are set according to the recommendations in (Li & Spratling, 2023).

## 3 Experiments

The above describes the default training protocol of our SimpleAT. Here, we introduce some other settings.

**Dataset and Network Setup.** We conduct experiments on three widely-used datasets, including CIFAR-10 (Krizhevsky et al., 2009), CIFAR-100 (Krizhevsky et al., 2009), SVHN (Netzer et al., 2011). In addition, we evaluated the model's robustness using CIFAR-10-C, which includes 15 types of corruption categorized into four groups: noise, blur, weather, and digital (Hendrycks & Dietterich, 2019). In most experiments, we use the ResNet-18 model (He et al., 2016) as the backbone. Moreover, we use WideResNet (Zagoruyko & Komodakis, 2016) as the backbone to evaluate the robust performance of CIFAR-10.

**Evaluation Protocol.** During the testing phase, we evaluate the accuracies on natural and adversarial images. To generate the adversarial images, we primarily use four attack methods: FGSM-attack, PGD-attack, C&W-attack, and AutoAttack (Croce & Hein, 2020b). We use common experimental settings for most experiments, where the perturbation budget is $\epsilon = 8/255$. For the PGD-attack and C&W-attack, we initialize the perturbation via a random Gaussian noise and then calculate the perturbation using Eq.2 and Eq.3 with 10-step iterations while $\alpha = 2/255$. For the FGSM-attack, we use the original FGSM-attack (as Eq.4 and Eq.5 shown), and we set $\alpha = 10/255$ and $\epsilon = 8/255$. For the AutoAttack, we follow the same setting in (Pang et al., 2021; Dong et al., 2022), which is composed of an ensemble of diverse attacks, including APGD-CE (Croce & Hein, 2020b), APGD-DLR (Croce & Hein, 2020b), FAB attack (Croce & Hein, 2020a), and Square attack (Andriushchenko et al., 2020).

### 3.1 Main Results

In this subsection, we ablate our SimpleAT through controlled experiments under PGD-AT paradigm. Several intriguing properties are observed.

**The First Glance of Square Loss.**

Figure 1 shows the influence of the original square loss (OSL)[3] and rescaled square loss (RSL). We conduct experiments on a classical protocol (Rice et al., 2020), which is widely used in most recent defense methods. We call this the *default protocol*. Note that this protocol differs from ours in two significant ways, *i.e.*, it employs cross-entropy loss (CEL) and a piece-wise decay schedule[4], while other factors are the same as ours. First, we simply replace CEL with OSL or RSL and show test accuracy curves and adversarial accuracy curves. The results are shown in Figure 1 (a) and (b).

---

[3]The original square loss is to calculate the Euclidean distance between softmax features and one-hot label vectors. And its formulation is defined as $L(\boldsymbol{x}, \boldsymbol{y}) = \|g(\boldsymbol{f}(\boldsymbol{x})) - \boldsymbol{y}_{hot}\|$, where $g(\cdot)$ is the softmax operation.

[4]Here, the piecewise decay schedule divides the learning rate by 10 at the 100-th epoch and 150-th epoch.

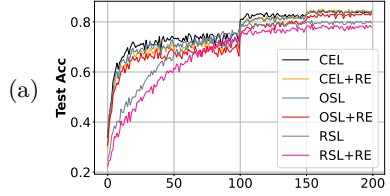 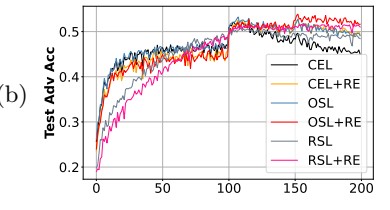

| | Nat-D | PGD-D | AA-B | AA-L | AA-D |
|---|---|---|---|---|---|
| CEL | 2.52 | 7.14 | 47.88 | 41.07 | 6.81 |
| CEL+RE | 3.18 | 3.45 | 47.55 | 45.52 | 2.03 |
| OSL | 4.40 | 4.13 | 48.43 | 44.73 | 3.70 |
| OSL+RE | 1.96 | 1.99 | **49.75** | **47.39** | 2.36 |
| RSL | 1.60 | 2.59 | 45.49 | 41.97 | 3.52 |
| RSL+RE | **0.64** | **0.74** | 46.04 | 44.82 | **1.22** |

(c) Quantitive results.

Figure 1: Analysis of square loss under the default protocol. All results are evaluated on a robust ResNet-18 trained on CIFAR-10. Subfigures (a) and (b) display the test accuracy curves, while the final one reports the quantitive results.

We observe that AT trained with CEL has a serious overfitting problem, where the test adversarial accuracy begins to drop after the first learning rate drop. But training a robust model under the supervision of OSL will reduce the risk of robust overfitting, compared to that trained with CEL. Then, we consider using a random erasing technique, which can enable us to take a closer look at the phenomenon of AT with OSL. We can find that, *by switching to square loss, both adversarial and natural accuracy curves are very close to the robust model trained with the combination of CEL and random erasing.* Furthermore, the combination of OSL and random erasing can achieve better results.

In addition, Figure 1 (c) reports the results that indicate the difference between the highest and final checkpoints. These results can quantitively analyze the robust overfitting. Here, we also report the adversarial accuracy under AutoAttack. According to the results, we find that a robust model trained with OSL actually achieves better results, and the gap between the highest and final checkpoints is smaller than that trained with CEL. In sum, AT trained with OSL achieve consistently better results when compared to the typical PGD-AT, and it can mitigate overfitting to some extent, which is equivalent to using data regularization on top of CEL. This echoes the existing results reported in (Hu et al., 2022; Pang et al., 2022). However, when compared to the previous work (Dong et al., 2022), Figure 1 (c) also tells us the fact that a simple replacement of the loss function does not lead to significant performance improvement. And the robust overfitting problem is not well addressed.

We further investigate the effect of RSL under this classical protocol. Unfortunately, the model cannot be well trained under RSL supervision using the default settings. We argue that the main problem is the large beginning learning rate, which is 0.1 as usual, leading to gradient explosion. To make the training available, we modified the beginning learning rate to 0.001 and divides it by 10 at the 100-th epoch and 150-th epoch. We show the results in Figure 1. We find that using RSL can achieve a smaller degradation in both robust and natural accuracy, indicating that *RSL is benefiting to mitigate robust overfitting.*

**Learning Rates.** An important design of our training protocol is the one-cycle learning rate. We investigate the impact of this design through controlled experiments, as shown in Figure 2 and Table 1. This design is necessary because the robust model trained with RSL remained sensitive to AutoAttack. We argue that modifying the learning rate schedule in the training protocol is crucial for achieving a robust model.

Figure 2 shows the influence of the one cyclic learning rate (OCL). First, subfigure (a) shows that using the OCL schedule leads to smooth training loss curves. Then, CEL+OCL can reduce the risk of robust overfitting significantly. Although the performance at the final checkpoint is slightly degraded, this is acceptable compared to previous experimental results. Third, both OSL+OCL and RSL+OCL can achieve the best performance at the final checkpoints, where robust overfitting is missing.

| | Natural | FGSM | PGD-10 | AA |
|---|---|---|---|---|
| CEL | 82.52 | 60.44 | 48.74 | 45.19 |
| OSL | 84.21 | 62.30 | 52.55 | 48.04 |
| RSL | 84.89 | 63.41 | 53.68 | 48.59 |

Table 1: Test accuracy (%) on CIFAR-10. Experiments are conducted at the final epoch.

Table 1 reports the quantitive results at the end of the training, when the OCL schedule is used. For CEL, the performance of the robust model against PGD-attack is on par with that of the model trained with a piecewise decay schedule (refer to CEL in Figure 1), but the adversarial accuracy under AutoAttack (*i.e.* 45.19%) is worse. Then, we find that simply replacing CEL with OSL can

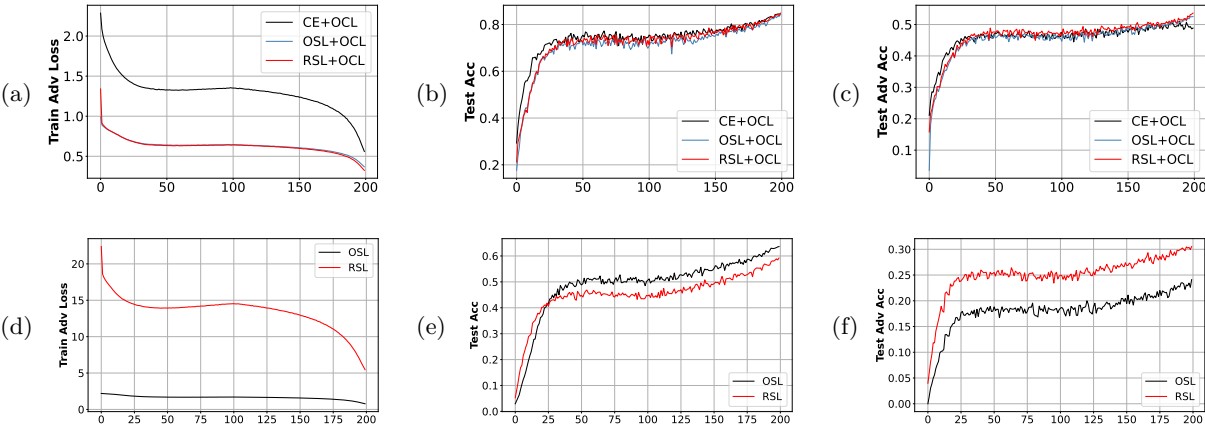

Figure 2: Analysis of learning rates. The results are based on a robust ResNet-18 model trained on CIFAR-10 and CIFAR-100. The top row displays the result curves for different loss functions under the same learning rate scheduler, while the bottom row focuses on studying the effect of OSL and RSL on CIFAR-100.

make the model achieve similar results (either natural accuracy or adversarial accuracy) that are reported by using default protocol. Finally, this table also shows that a robust model trained with RSL+OCL can achieve better results compared to OSL+OCL, while RSL also improves the performance when the input is a clean image.

Figure 2 (d-f) shows the result curves on CIFAR-100, where class number (*i.e.*, 100) is larger than 42. By using the OCL, robust overfitting is also missing. All the best results are achieved at the final epoch. Moreover, (Hui & Belkin, 2021) has mentioned that square loss always has a problem, therefore, rescaled square loss (RSL) is used. We observe a similar phenomenon in adversarial training, where the robust model trained with OSL achieves worse adversarial accuracy (*i.e.*, 24.11% under PGD-attack). Interestingly, when we use RSL, the performance gain is significant. For instance, the robust accuracy will be increased to 30.54% under PGD-attack. However, training under the supervision of RSL will decrease natural accuracy (as shown in Figure 2 (e)). We think that this performance degradation is relatively acceptable compared to robustness changes, due to the well-known trade-off phenomenon (Zhang et al., 2019).

Overall, *modifying the training protocols to include square loss and one cyclic learning rate is all we needed.* This allows us to achieve good results, even with a large number of classes, while also neglecting the disreputable overfitting problem.

**Data Augmentation.** In Figure 3, we study the influence of data augmentation in our training protocol. Specifically, we compare two different data augmentation methods, both of which contain random erasing. The first default method utilizes RandomCrop, RandomHorizontalFlip, and random erasing. And the second one is a recent state-of-the-art method, called IDBH, which consists of CropShift, ColorShape, and random erasing. We follow the default setting recommended by Li & Spratling (2023).

Figure 3 (a) and (b) show the results, when random erasing is used or not used in IDBH. First, we observe that all methods perform best during the final phase of training. Where robust overfitting is mitigated well. Second, whether random erasing is used or not, the difference in natural accuracy between different methods is not very large. On the other hand, subfigure (b) shows that using random erasing improves robust accuracy for both CEL and RSL. But, when using CEL and random erasing, we find that the robust accuracy has a little degradation at the final checkpoints, although it is better than that trained without random erasing. This phenomenon does not exist when we use RSL as the supervisor. That is, using both RSL and random erasing can not only achieve good results but also reach a good trade-off between accuracy and robustness. Because previous works show that robustness improvement will bring accuracy drop, but our observation shows that ours can improve robustness without accuracy degradation.

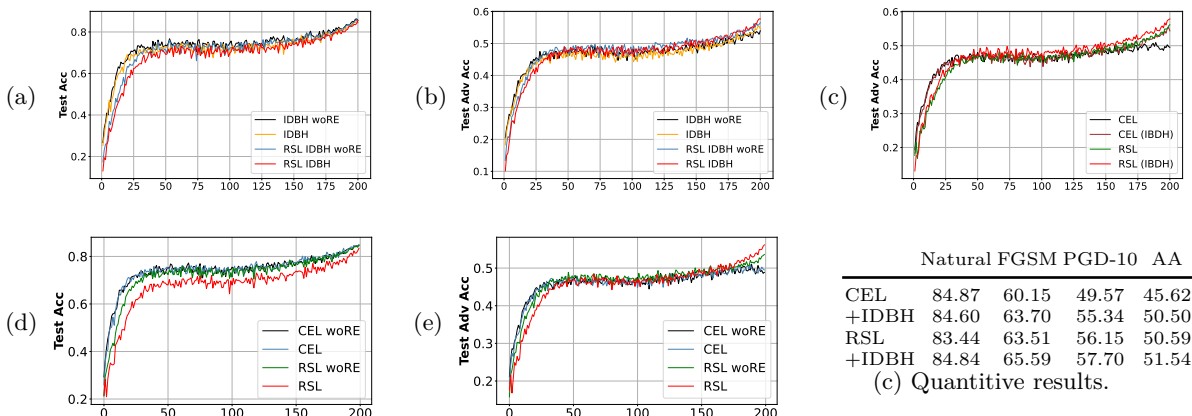

Figure 3: Analysis of random erasing. All results are evaluated on a robust ResNet-18 trained on CIFAR-10. Subfigure (a) and (b) show the test accuracy curves when we study the influence of random erasing modules in the IDBH scheme. Subfigure (c) study the effect of the IDBH. And Subfigure (d) and (e) displays the effect of the default data augmentation with/without using random erasing. The final subfigure reports the quantitive results.

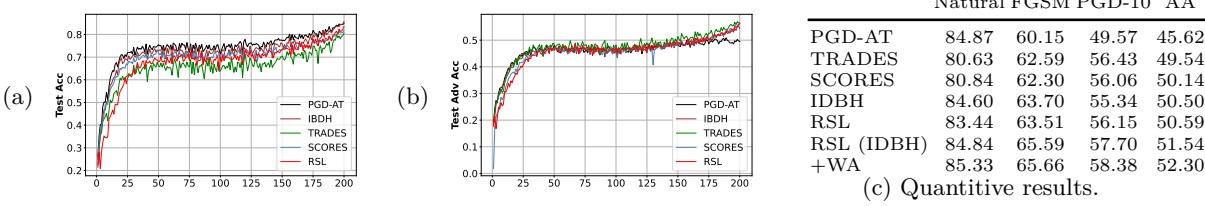

Figure 4: Further analysis. All results are evaluated on a robust ResNet-18 trained on CIFAR-10. Subfigure (a) shows the clean accuracy curves, subfigure (b) draws the adversarial accuracy curves, and the final subfigure reports the quantitive results.

Moreover, we draw the result curves under the default data augmentation in Figure 3 (d) and (e). We also observe a similar phenomenon that the best performance is achieved at the final phase of training. But the robust model trained with CEL is unstable at this phase, with even minor performance degradation. In subfigure (e), we find that using RSL without random erasing can also achieve good robustness while maintaining accuracy on par with a model trained with both CEL and random erasing. Although RSL together with random erasing has a little accuracy degradation, it can achieve the highest robust accuracy. When compared to subfigures (a) and (b), we conclude that IDBH is a better data augmentation for learning a robust model, which not only improves the robustness but also maintains satisfactory accuracy. Our quantitive results reported in Table 3 also echo these observations. In the following experiments, we mainly use IDBH as our data augmentation method, which helps us to achieve good results.

Overall, combing the random erasing and OCL also mitigates robust overfitting under CEL for standard AT. Therefore, *modifying training protocols to include random erasing and one cyclic learning rate is all that is needed.*

**Further Analysis.** In Figure 4, we compare RSL-based AT with other four representative methods, *i.e.*, PGD-AT (Rice et al., 2020), TRADES (Zhang et al., 2019), SCORES (Pang et al., 2022), and IBDH (Li & Spratling, 2023). All compared methods used the same training protocols as previously highlighted, including one cyclic learning rate and random erasing. The primary difference among the methods was the training objective, with PGD-AT and IBDH using cross-entropy loss as the default training objective. TRADES is a classical baseline that usually achieves competitive results. SCORES, a recent state-of-the-art method, demonstrates that replacing the KL-divergence with Euclidean distance can achieve good results.

| Method | Protocol | Natural | | | PGD-10 | | | AutoAttack | | |
|---|---|---|---|---|---|---|---|---|---|---|
| | | Best | Final | Diff | Best | Final | Diff | Best | Final | Diff |
| PGD-AT | Default | 81.48 | 84.00 | 2.52 | 52.06 | 44.92 | 7.14 | 47.88 | 41.07 | 6.81 |
| SAT | | 82.81 | 81.86 | 0.95 | 53.81 | 53.31 | 0.50 | 50.21 | 49.73 | 0.48 |
| KD-SWA | | 84.84 | 85.26 | 0.42 | 54.89 | 53.80 | 1.09 | 50.42 | 49.83 | 0.59 |
| Co-teaching | | 81.94 | 82.22 | 0.28 | 51.27 | 50.52 | 0.75 | 49.60 | 48.49 | 1.11 |
| TE | | 82.35 | 82.79 | 0.44 | 55.79 | 54.83 | 0.96 | 50.59 | 49.62 | 0.97 |
| SRC | | 80.30 | 80.70 | 0.40 | 57.55 | 57.90 | 0.35 | 50.38 | 50.35 | 0.03 |
| TRADES | Ours | 79.17 | 80.63 | 1.46 | 56.80 | 56.43 | 0.46 | 49.07 | 49.54 | 0.47 |
| SCORES | | 80.84 | 80.84 | 0.00 | 56.06 | 56.06 | 0.00 | 50.14 | 50.14 | 0.00 |
| IDBH | | 84.64 | 85.59 | 0.95 | 55.39 | 54.61 | 0.78 | 50.66 | 50.00 | 0.66 |
| SimpleAT | | 83.58 | 83.58 | 0.00 | 56.22 | 56.22 | 0.00 | 50.61 | 50.61 | 0.00 |
| SimpleAT (IDBH) | | 84.84 | 84.84 | 0.00 | 57.70 | 57.70 | 0.00 | 51.54 | 51.54 | 0.00 |
| SimpleAT (IDBH+WA) | | 85.33 | 85.33 | 0.00 | 58.38 | 58.38 | 0.00 | 52.30 | 52.30 | 0.00 |

Table 2: Analysis of robust overfitting. Test accuracy (%) of several models trained on CIFAR-10. The backbone we used here is ResNet-18. The "Natural" is the natural accuracy. The "Best" reports the results of the checkpoint and who achieves the best performance during the training. The "Final" is the results of the last training epoch. The "Diff" shows the difference between the best and final checkpoints.

Moreover, except for the default data augmentation, we use IDBH in our RSL-based AT. And following recent works (Rebuffi et al., 2021b; Pang et al., 2022), we also use the weight averaging technique (WA) to further enhance robustness and accuracy.

Subfigures (a) and (b) show the accuracy and robustness curves on test data. We observe that all compared methods achieved their best performance at the final checkpoints when using one cyclic learning rate and random erasing. This finding differs from previous observations in (Rice et al., 2020), where robust over-fitting is consistently observed when using these techniques separately. Moreover, we observe that using regularization techniques, like TRADES and SCORES, can improve model robustness, but the accuracy on natural data has degradations, as shown in Figure 4 (a). Interestingly, simply using RSL as the training objective can improve robustness but the accuracy drop is not significant. The trade-off problem between robustness and accuracy can be alleviated.

Figure 4 reports the quantitive results on the final checkpoints for all compared methods. We find that the performance of RSL-based AT is the best overall. It achieves the best defense against even the strongest attack AutoAttack. Then, we change our default data augmentation to the strong IDBH method, as the last second row is shown in Figure 4. Our proposed protocols can achieve the best performance for both robustness and accuracy. Finally, by using the WA technique, the overall results are further improved and very good results are achieved. For example, our method can achieve 52.3% robust performance against AutoAttack.

## 3.2 Comparisons with Previous Works

In the previous subsection, we conducted detailed ablation studies using controlled experiments. The results showed that our proposed training protocol is competitive when compared to previous default protocols. In this subsection, we will compare the results with those of many recent state-of-the-art methods under the PGD-based AT paradigm. It's worth noting that our robust model is trained only under the proposed training protocol without adding other regularizations or modifying the architectures.

We conduct an analysis of the problem of robust overfitting for various methods, including SAT (Huang et al., 2020), KD-SWA (Chen et al., 2021), TE (Dong et al., 2022) and SRC (Liu et al., 2023), which aim to mitigate robust overfitting. Additionally, we compare various regularization and data augmentation methods, such as TRADES (Zhang et al., 2019), SCORES (Pang et al., 2022) and IDBH (Li & Spratling, 2023), trained under our proposed training protocol.

| Method | TRADES | ES | SAT | FAT | LBGAT | AT-HE | Tricks | LTD | IDBH | SimpleAT | SimpleAT (+WA) |
|---|---|---|---|---|---|---|---|---|---|---|---|
| Natural | 84.92 | 85.35 | 83.48 | 84.52 | 88.70 | 85.14 | 86.43 | 85.02 | 89.93 | 90.04 | 89.82 |
| AutoAttack | 53.08 | 53.42 | 53.34 | 53.51 | 53.57 | 53.74 | 54.39 | 54.45 | 54.10 | 54.55 | 55.95 |

Table 3: Performance of several models trained on CIFAR-10. The backbone is WideResNet.

| | ES | TRADES | ARD | RSLAD | AT-TE | SimpleAT | SimpleAT (IDBH) |
|---|---|---|---|---|---|---|---|
| Natural | 57.74 | 58.67 | 60.79 | 57.83 | 57.12 | 57.19 | 60.89 |
| PGD | 29.27 | 29.88 | 28.11 | 30.55 | 30.24 | 31.50 | 34.60 |
| AutoAttack | 25.01 | 25.27 | 24.68 | 26.41 | 25.34 | 25.14 | 27.89 |

(a) The evaluation results on CIFAR-100.

| | ES | AT+TE | TRADES+TE | SimpleAT |
|---|---|---|---|---|
| Natural | 89.00 | 90.91 | 88.52 | 91.11 |
| PGD | 54.51 | 59.05 | 58.49 | 61.10 |
| AutoAttack | 46.61 | 50.61 | 50.16 | 52.10 |

(b) The evaluation results on SVHN.

Table 4: Performance of several models trained on CIFAR-100 and SVHN. The backbone is ResNet-18.

Table 2 presents the results on CIFAR-10, showing that all methods, except for the *de-facto* adversarial training (PGD-AT), perform well in reducing the risk of robust overfitting. Under our training protocol, we observe that classical TRADES can achieve competitive results in the final stage of training when compared to more recent techniques such as TE and SRC. Then, by simply replacing the KL divergence with the Euclidean distance, the SCORES can further improve the performance, and the robust overfitting problem is well mitigated. Although both trading and scoring exhibit good robustness, this is achieved at the expense of a certain degree of natural precision, namely facing a trade-off dilemma. Interestingly, our SimpleAT achieves the best performance at final checkpoints and maintains a good balance between natural and adversarial accuracy, surpassing all compared methods. Moreover, the utilization of IDBH and weight averaging techniques leads to a significant enhancement in overall performance. Our approach surpasses most cutting-edge methods, as evidenced by the last row of Table 2. The adversarial accuracy against AutoAttack reaches 52.30%, which is a notable result on the leaderboard (when the backbone is ResNet-18).

We also evaluate the performance when using a larger network, *i.e.*, WideResNet, in which we compare our SimpleAT with various recent robust models, including SAT (Huang et al., 2020), FAT (Zhang et al., 2020), LBGAT (Cui et al., 2021), AT-HE (Pang et al., 2020b), Tricks (Pang et al., 2021), LTD (Chen & Lee, 2021), IDBH (Li & Spratling, 2023). Similar to (Rice et al., 2020), we also use an early-stopping strategy (ES) to avoid robust overfitting, which can also achieve good results. The results are reported in Table 3. We find that our SimpleAT consistently outperforms all the compared methods in terms of both natural accuracy and robust performance. Echoing the results on ResNet-18, the robustness can be further improved by using weight averaging techniques. Although there is a slight degradation in the natural accuracy, it is competitive since the 89.82% score is better than most of the compared methods and only lower than IDBH.

Moreover, we report the results on CIFAR-100 in Table 4 (a), where we add two competitive distillation-based methods, such as ARD (Goldblum et al., 2020) and RSLAD (Zi et al., 2021). Our SimpleAT achieves competitive results. Although our SimpleAT outperforms RSLAD against PGD-attack, it underperforms in natural accuracy and strong AutoAttack. We think this is attributed to the fact that RSLAD employs a complex adversarial distillation technique, while our SimpleAT follows a simpler training protocol. When transferring the random erasing to strong IDBH, SimpleAT (IDBH) achieves the best performance on both accuracy and robustness, where the robust performance against AutoAttack has a gain of 5.6%, compared with RSLAD.

The results on SVHN are presented in Table 4 (b), where we compare our method with two TE methods, namely AT+TE and TRADES+AT, both of which achieve state-of-the-art performance. Notably, we use only 30 epochs to train our robust model. This modification is due to the fact that the dataset is relatively simple and the robust models are always easier to train while achieving higher accuracy. The results in

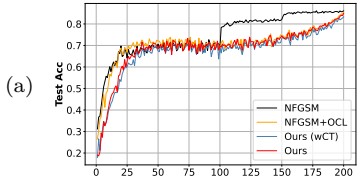
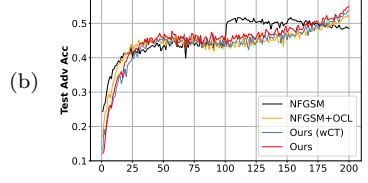

| | NFGSM | PGD |
|---|---|---|
| Natural | 60.74 | 60.89 |
| FGSM | 39.46 | 40.51 |
| PGD-10 | 32.86 | 34.60 |
| AA | 26.57 | 27.89 |
| Time (h) | 2.03 | 10.34 |

(c) Performance on CIFAR-100.

Figure 5: Analysis of FGSM-based AT methods. All models are trained on CIFAR-10/100. The backbone is ResNet-18. Subfigure (a) shows the clean accuracy curves, subfigure (b) draws the adversarial accuracy curves, and the final subfigure reports the quantitive results.

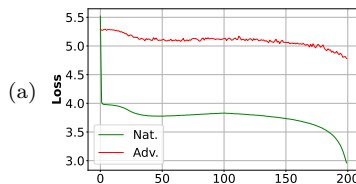
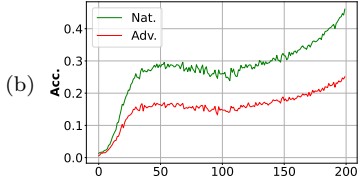

| Method | Natural | PGD | AA |
|---|---|---|---|
| FGSM | 41.37 | 17.05 | 12.31 |
| AT-GA | 45.52 | 20.39 | 16.25 |
| FAST-BAT | 45.80 | 21.97 | 17.64 |
| Ours | 46.08 | 25.05 | 18.68 |

(c) Quantitive results.

Figure 6: Compared results of different FGSM-based methods trained on Tiny-ImageNet. The backbone is ResNet-18. Subfigure (a) shows the learning loss curves, subfigure (b) draws the test accuracy curves, and the final subfigure reports the quantitive results.

this table demonstrate that SimpleAT again achieves the best performance while achieving a good balance between accuracy and robustness. In general, SimpleAT helps improve both natural and adversarial accuracy.

### 3.3  Compared with FGSM-based AT methods

In this subsection, we mainly evaluate the performance under a more effective FGSM-based AT paradigm. We compare three baseline methods, such as AT-GA (Andriushchenko & Flammarion, 2020), FAST-BAT (Zhang et al., 2021b), and NFGSM (de Jorge et al., 2022). We report our results in Figure 5 and Figure 6.

In Figure 5 (a) and (b), we study whether SimpleAT suffers from catastrophic overfitting. Echoing the results shown in (de Jorge et al., 2022), NFGSM indeed makes the adversarial training stable, but it still faces robust overfitting under such a longer training scheme. Therefore, we change the learning rate schedule to one cyclic learning rate (OCL). The risk of robust overfitting can be reduced, similar to PGD-based AT, as Figure 5 (b) shows. After comparing the yellow and blue curves in Figure 5, we find that RSL improves the robustness but slightly degrades natural accuracy. Based on the previous experience, we further use IDBH for the data augmentation, resulting in improved natural accuracy that is comparable to the CEL-trained model. Moreover, we also find that the robust accuracy against PGD-10 attack is 54.77%, which is comparable to that model trained with PGD-based AT. To further verify this, we conduct similar experiments on CIFAR-100 and report the results in Figure 5 (c). A similar phenomenon is observed: SimpleAT is effective and performs similar results to PGD-based AT. It's worth noting that PGD-based AT requires 5 times more training time. That is, SimpleAT

Figure 6 shows the results on a large real-world dataset, i.e., Tiny-ImageNet, where we compare SimpleAT with three high-effect AT methods, i.e., FGSM, AT-GA, and FAST-BAT. Note that the original IDBH does not provide recommended hyper-parameter settings for Tiny-ImageNet, so we use RandomCrop, RandomHorizontalFlip, and random erasing as our data augmentation scheme. Figure 6 (a) displays the learning loss curves for natural and adversarial images, while Figure 6 (b) shows the test accuracy curves during training. Our SimpleAT method ensures stability in FGSM-based AT by preventing both robust and catastrophic overfitting. Furthermore, we show the quantitive results in Figure 6 (c). Our SimpleAT is consistently better than all compared methods. Specially, compared to FAST-BAT, SimpleAT reaches gains of 0.6%, 14.02%, and 12.24% on natural, PGD-50, and AutoAttack, respectively. That is, SimpleAT is able

| Arch | Method | Generated | Batch | Epoch | Clean | AutoAttack |
|------|--------|-----------|-------|-------|-------|-----------|
| WRN-28-10 | (Rebuffi et al., 2021b) | 1M | 1024 | 800 | 87.33 | 60.73 |
| | (Pang et al., 2022) | 1M | 512 | 400 | 88.10 | 61.51 |
| | (Wang et al., 2023) | 3.5M | 512 | 400 | 90.53 | 61.44 |
| | SimpleAT | 3.5M | 512 | 400 | 91.81 | 61.95 |

(a) Compared with previous works.

| Arch | Eval. | CEL | SL | SCORES | RSL(2,1) | RSL(1,1) | RSL(2,2) | RSL(3,1) | RSL(4,1) | RSL(5,1) |
|------|-------|-----|-----|--------|----------|----------|----------|----------|----------|----------|
| WRN-28-10 | Natural | 92.05 | 91.35 | 90.53 | 91.81 | 91.39 | 91.82 | 91.99 | 92.01 | 92.07 |
| | PGD | 67.25 | 68.16 | 69.34 | 69.34 | 68.86 | 68.75 | 69.24 | 68.85 | 68.95 |
| | AutoAttack | 61.79 | 61.43 | 61.44 | 61.95 | 61.51 | 61.87 | 61.72 | 61.82 | 61.47 |
| WRN-76-16 | Natural | 93.45 | 92.44 | - | 93.08 | - | | | | |
| | AutoAttack | 65.33 | 65.32 | - | 65.62 | - | | | | |

(b) Compared with different loss functions.

Table 5: Test accuracy (%) on CIFAR-10 when using extra dataset during adversarial training. In subtable (b), all methods are trained by using the same 3.5 million generated images for a fair comparison. RSL(2,1) means that we set $k = 2$ and $M = 1$.

to substantially improve robustness while maintaining high natural accuracy. The results further verify our claim that SimpleAT yields a favorable balance between adversarial and natural accuracy.

## 3.4 Compared with semi-supervised methods

Recently, researchers use extra datasets (real or synthetic) to enhance model robustness, resulting in top-tier performance on the RobustBench (Rebuffi et al., 2021b; Pang et al., 2022; Wang et al., 2023). We adopt the experimental setup from (Wang et al., 2023), which employs a state-of-the-art diffusion model to generate 3.5 million images sampled from a larger pool of 5 million virtual images. We default to using WideResNet-28-10 as the backbone. See Table 5 (a). Our SimpleAT performs better with additional data for training, achieving top-1 robustness against AutoAttack and significantly improving natural accuracy. For instance, SimpleAT shows a clean data accuracy gain of about 1.43% when compared to (Wang et al., 2023).

Moreover, we study the effect of different training objectives, in which we select the cross-entropy loss, a recent Squentropy loss (SL)(Hui et al., 2023), the SCORES objective, and our mentioned rescaled square loss. The results are shown in Table 5 (b). We first find that the natural accuracy grows as the value of $k$ increases. But the robust performance achieves the best when $k = 2$. Moreover, when we increase the parameter $M$ to 2, both natural and robust accuracy will be degraded. So, we default to using k=2 and M=1 as our default settings in this experiment. Compared to CEL and SQE, RSL is more effective in defending against adversarial attacks. However, the natural accuracy is slightly lower than that achieved with CEL during training. Finally, we evaluate performance using a larger backbone - WideResNet-76-16. SimpleAT can reach 65.53% adversarial accuracy against AutoAttack and 93.08%. This means that SimpleAT can achieve the top-rank performance on the RobustBench leaderboard. However, due to our limited computational resources, we are not able to further scale up the data amount to enhance robustness, which may take months to train one model. But we believe that our SimpleAT can achieve better results when using the larger extra dataset (like with 50M images).

## 3.5 Experiments on CIFAR-10-C

All foregoing experiments are conducted under adversarial attacks, which mainly reflect the worst-case performance for an evaluated model. However, in the real world, there are so many other types of corruption

| Method | Standard | 100% Gauss | 50% Gauss | Fast PAT | AdvProp | $l_\infty$ adv. | $l_2$ adv. | RLAT | SimpleAT |
|--------|----------|-----------|-----------|----------|---------|-----------------|-----------|------|----------|
| Standard Acc. | 95.1 | 92.5 | 93.2 | 93.4 | 94.7 | 93.3 | 93.6 | 93.1 | 94.0 |
| Corruption Acc. | 74.6 | 80.5 | 85.0 | 80.6 | 82.9 | 82.7 | 83.4 | 84.1 | 87.2 |

Table 6: Evaluation on CIFAR-10-C. We report the accuracy of ResNet-18 models trained on CIFAR-10.

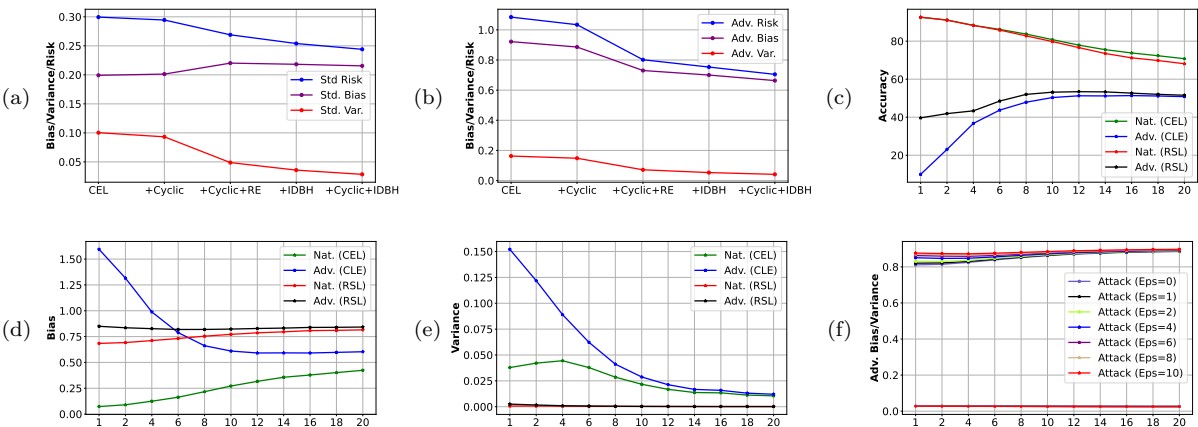

Figure 7: Bias, variance, and risk for $l_\infty$ adversarially trained models on the CIFAR10 using ResNet18.

that affect the model performance, like noise, blur, digital, and weather corruption. To further verify the effectiveness of our method, we conduct experiments on CIFAR-10-C. Following recent results in (Kireev et al., 2022), we use a small perturbation budget where $\epsilon = 2/255$. And we calculate the average accuracy over 15 different common corruptions, which show the average-case behavior of the robust model. We compare eight different baseline methods, including standard training, $l_2$ and $l_\infty$ adversarial training, Gaussian augmentation (with both 50% and 100%), AdvProp (Xie et al., 2020), PAT (Laidlaw et al., 2021), and RLAT (Kireev et al., 2022). See Table 6. SimpleAT also achieves the best robust performance, which has a 3.69% gain compared to the second-best RLAT. Although the standard accuracy is not the best, we think our result of 94.0% is also competitive.

## 4 Further Analysis

### 4.1 Bias-variance decomposition

We begin our analysis by exploiting well-known bias-variance decomposition. Following (Yu et al., 2021), the adversarial expected risk can be decomposed as:

$$\mathbb{E}_{x,y}\mathbb{E}_{\mathcal{T}}\big[\|y - f(x, \mathcal{T})\|\big] = \mathbb{E}_{x,y}\big[\|y - \hat{f}(x)\|\big] + \mathbb{E}_x\mathbb{E}_{\mathcal{T}}\big[\|f(x, \mathcal{T}) - \hat{f}(x)\|\big], \tag{9}$$

where $\hat{f}(x) = \mathbb{E}_{\mathcal{T}} f(x, \mathcal{T})$, and $\mathbb{E}_{\mathcal{T}}\big[\|y - f(x, \mathcal{T})\|\big]$ measures the average prediction error over different realizations of the training samples. As a result, we uniformly separate the whole training data into two parts $\{\mathcal{T}_1, \mathcal{T}_2\}$, and use the training protocol highlighted in our SimpleAT. The results are shown in Figure 7. Here we refer to the bias and variance of natural images as "natural bias and variance," while those of adversarial examples are referred to as "adversarial bias and variance"

We investigate the impact of one cyclic learning rate and erasing-based data augmentation, as shown in Figure 7 (a) and (b). Our findings reveal that natural variance dominates natural empirical risk, but adversarial bias remains the primary factor affecting adversarial empirical risk since adversarial variance is low. These two figures also show that data augmentation can decrease both natural and adversarial variances, leading to well robustness and accuracy improvement. Moreover, our observations confirm previous claims in (Smith, 2017) and (Chen et al., 2020b), because they mentioned that OCL or data augmentation can reduce the variance of a trained model, respectively. Interestingly, (Yu et al., 2021) claimed that "adversarial training increases the variance of the model predictions and thus leads to overfitting", which verifies our result, *i.e.*, combing RE and OCL makes the robust model achieve lower variance while mitigating robust overfitting.

We compare the difference between CEL and RSL. See Figure 7 (c-e). We train our robust model under different perturbation budgets, *e.g.*, $\epsilon$ is from 1 to 20. And all the adversarial results are evaluated under

the PGD-attack with 10 steps and $\epsilon = 8$. Figure 7 (c) shows that RSL-trained models consistently achieve superior adversarial accuracy compared to CEL-trained models, albeit with slightly lower natural accuracy.

We compare the results of natural and adversarial bias/variance. See Figure 7 (d) and (e). For CEL-trained models, the natural bias increases monotonically, and the adversarial bias decreases rapidly under a small perturbation budget and then flattens out. The natural variance curve of the CEL-trained model is unimodal which is similar to the phenomenon observed by Yu et al. (2021), and the adversarial variance curve is similar to the corresponding natural bias curve. On the other hand, RSL-trained models have a very different tendency of curves. Where both natural and adversarial bias changes a little with changing the perturbation budgets, and two variance scores of RSL-trained models are extremely low. Moreover, with small perturbation budgets, both adversarial bias and variance of CEL-trained models are totally higher. The gaps between natural and adversarial bias/variance are also large. This means the models are easily attacked. Then, with the increasing perturbation budget, the gaps become smaller, and the adversarial bias and variance are also becoming small enough. It means that the model becomes robust against adversarial attacks.

The aforementioned observations reveal intriguing findings. Specifically, with small perturbation budgets, models trained with CEL exhibit significantly higher adversarial bias and variance. The gap between natural and adversarial bias/variance is substantial, indicating that these models are highly vulnerable to attacks. However, as the magnitude of perturbation increases, the gaps gradually narrow and both adversarial bias and variance decrease correspondingly. This suggests that the models become more robust against adversarial attacks. Moreover, for a model trained with RSL, the difference in bias and variance between natural and adversarial examples is minimal, accompanied by smaller variances. It can be inferred that *a robust model should exhibit a small gap between natural and adversarial bias while maintaining lower variances.*

Finally, the results for SimpleAT are summarized in Figure 7 (f), which shows similar tendency curves with varying perturbation size $\epsilon$. This observation further supports our inference, which indicates that our SimpleAT indeed achieves good robustness and accuracy. Furthermore, we observe that the choice of loss function impacts the final bias and variance as shown in Figure 7. We believe that the current methods of adversarial attack optimize a loss function with a formulation similar to that of the adversarial bias used for learning perturbation noise. It is worth noting that rescaled square loss function is closely related to the definition of adversarial bias. As a result, we revisit the rescaled square loss function and provide a more in-depth analysis in the next subsection.

### 4.2 Relations to Logit Penalty Methods

We finally provide an in-depth analysis based on the RSL, by introducing a perspective as:

**Perspective 1.** RSL adaptively constrains logit features to be small in $L_2$ ball.

We rewrite the formulation of RSL as follows:

$$L(\boldsymbol{x}, \boldsymbol{y}) = \|\boldsymbol{k} \cdot (\boldsymbol{f}(\boldsymbol{x}) - M \cdot \boldsymbol{y}_{hot})\|_2^2 \sim \boldsymbol{a}\|\boldsymbol{f}(\boldsymbol{x})\|_2^2 - 2\boldsymbol{b}\boldsymbol{f}^T(\boldsymbol{x})\boldsymbol{y}_{hot}, \tag{10}$$

where $\boldsymbol{a} = \|\boldsymbol{k}\|$ and $\boldsymbol{b} = 2M\|\boldsymbol{k}\|$. These two parameters remain constant when fixed during training. In fact, RSL comprises two distinct functions: (i) the first one is to minimize the $L_2$-norm of the logit feature, *i.e.*, $\boldsymbol{f}(\boldsymbol{x})$; (ii) the second one is equal to the function of CEL (without the softmax operation). Therefore, we argue that the primary distinction from CEL lies within the first item and *the effect of RSL is due to its adaptive constraint on the L2-norm of logit features.*

Recent works in (Stutz et al., 2020; Wei et al., 2022; Liu et al., 2023) have shown that overconfidence usually leads to overfitting or worse generalization. Therefore, recent work usually considers using logit penalty, logit normalization, and label smoothing. Logit penalty (LP) (Kornblith et al., 2021) is to impose a penalty on the logit features, which can be formulated as $L_{LP} = L(\boldsymbol{x}, \boldsymbol{y}) + \beta\|\boldsymbol{f}(\boldsymbol{x})\|_2^2$, where $\beta$ is the hyperparameter. Logit normalization (LN) (Wei et al., 2022) is to replace the original logit features with its $L_2$ normalization features, *i.e.*, $\hat{\boldsymbol{f}} = \boldsymbol{f}/(\tau\|\boldsymbol{f}\|)_2^2$. Furthermore, label smoothing (LS) has been extensively employed in prior research, which has indeed facilitated improvements in results as demonstrated by Pang et al. (2021).

(a) 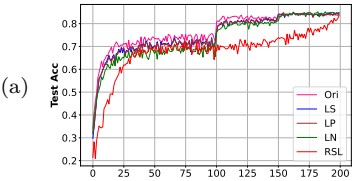

(b) 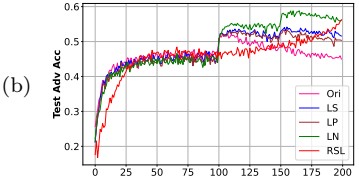

| Method | Natural | PGD | AA |
|---|---|---|---|
| LS | 82.69 | 55.26 | 49.75 |
| LP | 80.41 | 54.11 | 50.24 |
| LN | 80.23 | 58.80 | 46.38 |
| SimpleAT | 83.44 | 56.15 | 50.59 |

(c) Quantitive results.

Figure 8: Compared results of logit penalty methods trained on CIFAR-10. The backbone is ResNet-18. Subfigures (a) and (b) show the test accuracy curves and the final table reports the quantitive results.

Based on these, we further conduct experiments on CIFAR-10 with ResNet-18. The results are shown in Figure 8. We find that all logit penalty methods can mitigate overfitting problems and also achieve good results improvements. These results echo previous observations in (Stutz et al., 2020; Wei et al., 2022; Liu et al., 2023). That is, to achieve a good robust model, the confidence level needs to be suppressed. LN achieves impressive robust accuracy against PGD-attack but remains vulnerable to strong AutoAttack. We believe that adjusting the hyperparameters such as $\beta$ or $\tau$ in logit penalty methods is challenging. While SimpleAT also has two additional parameters ($k$ and $M$), they are easier to search for compared to finding the optimal hyperparameter of the regularization scheme. In general, SimpleAT can achieve the overall best performance, due to its implicit logit constraints.

## 5 Conclusion

We demonstrate how training protocols have a significant impact on model robustness. When controlling for various factors, we find that simple modifications to the prior training protocol can enhance model robustness. To this end, we introduce SimpleAT, a simple yet effective method that achieves a good trade-off between accuracy and robustness while mitigating robust overfitting. Our experiments have produced competitive results across various datasets, and our analysis also provides insights into the rationale behind these modifications. In the future, we aim to expand our analysis to include theoretical certifications.

### Broader Impact Statement

The defense against adversarial examples has garnered significant attention in the machine learning community due to the potential risks it poses to real-world applications. Therefore, it is very important to develop a robust model which can robust against different adversarial attacks. In this paper, we study the effect of training protocol, and we propose a simple but effective method that can improve the model's robustness and accuracy. We hope that it will help in building more secure yet simple models for real-world applications.

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
