# OpenReview forum: "Rethinking Adversarial Training with A Simple Baseline"
_TMLR — Withdrawn by Authors_

### Review · Reviewer_AkMt · 2023-07-03

**Summary Of Contributions:**

The authors propose a “simple” adversarial training procedure that is argued to outperform many alternative hyper-parameter settings and/or training procedure and adversarial training variants.

**Audience:**

Yes

**Broader Impact Concerns:**

No concerns.

**Claims And Evidence:**

No

**Requested Changes:**

See above.

**Strengths And Weaknesses:**

Strengths:
- Intriguing premise to improve performance just with a loss and learning rate change.
- Trying the squared loss for AT is to the best of my knowledge an interesting and open question.
- Generally good results with a ResNet-18 for adversarial robustness.
- Quite detailed experiments in terms of settings and quantity.
- Additional promising results on CIFAR10-C.

Weaknesses:
The main issue I see with this paper is that I was unable to really trace the individual protocols/improvements throughout the paper. Thus, I am not really convinced that this simple protocol actually improves results – despite the fact that the right baseline is unclear. This is the result of many individual difficulties I had following the paper:
- The main section is a bit too short and leaves several specific unclear in my opinion. This is problematic, as the paper *is* about these specifics. Examples:
-- What exact random erasing strategy is used and how is it different from others?
-- How exactly is the cyclic learning rate implemented? Usually there is a cycle (say 30 or 60 epochs) and we run X cycles. Here, the cycle is unclear, but 200 epochs are used (also see [b] below).
-- I am missing a convincing argument/experiment making the connection between overfitting and cross-entropy loss.
-- It is also confusing that additional data (in a semi-supervised setting) and data augmentation (= same training data but different transformations) is used somewhat interchangeably.
- While the method seems to focus on simple approaches (changing loss, learning rate schedule, data augmentation), hyper-parameters are seemingly chosen from prior work (e.g., for data augmentation in Note 3 (for the cyclic learning rate details are unclear, see above). This also affects baselines (with the authors stating that RSL with the “classical” protocol does not work due to a high learning rate. In all these cases, was there any hyper-parameter selection/optimization performed that could be presented as part of an ablation?
- Results are also presented inconsistently. Sometimes, early stopping is used and the difference best – last is included, sometimes not. As captions are often short, this makes it hard to track improvements. Table 1 and Figure 1 results are, for example not consistent (AA adv acc differing by ~4% at the last epoch). In this regard, the training/test curves are, in my honest opinion, actually distracting. They are useful to judge overfitting, but comparing actual performance is difficult. Similarly, the first few experiments consider CEL or RSL loss with and without random erasing (RE), but later experiments only consider “standard” (incl. RE) vs. IBDH.
- In Figure 2, I am also surprised how smooth the training curves with OSL are. There is clearly some processing in the curves as the test curves look as if they were evaluated each epoch, while the training curves do not.
- In Figure 3, it seems that training did not finish or at least it is unclear what happens when training >200 epochs as adv test accuracy clearly goes up towards the end.
- In the table in Figure 4 and Table 2, I am also confused by the baselines. What is the difference between “our” protocol IDBH and SimpleAT with IDBH (is it the cyclic learning rate or the square loss?). There is also no WA baseline and given the good performance of TE, it would also be interesting to see this evaluated with the proposed protocol.
- In Tables 3 and 4, there is only a small overlap between methods. LTD works well on CIFAR10, so it would be interesting to see this for CIFAR-100, as well.
- In the bias variance analysis I think evaluation with AA would be more appropriate. SimpleAT with RSL and an eps of <= 2 should not get as high adversarial accuracy. I feel this also impacts the variance/bias results significantly.
- In terms of results, the method seems to improve, but I feel the right baselines are usually missing (see above) and improvements might be smaller than presented. In Figure 1, RE has the largest effect, not RSL; in Table 1, the best epoch is not reported; in Figure 4, a WA baseline is missing (IDBH + WA could be closer); in Table 2, a WA baseline I smissing, the difference IDBH in our protocol and SimpleAT (IDBH) is unclear and the effect of the proposed protocl on TE or KD-SWA would be interesting; in Table 3, again, a WA baseline is missing, without which performance is very close to LTD and we would need a standard deviation to really make a statistically significant decision; in Table 4, RSLAD is better than SimpleAT, but not with IDBH – IDBH is not listed alone as a baseline though; …

Besides these points, I want to highlight that both [a] and [b] are very relevant
- [a] (similar to Peng et al.) looks at various settings in order to improve robustness.
- [b] should be discussed as it includes experiments with heavy data augmentation, label smoothing and Adam – all aspects the authors discuss at various points.

[a]  https://arxiv.org/pdf/2010.03593.pdf
[b]  https://arxiv.org/pdf/2104.04448.pdf

Minor things:
- There is a missing reference in the outline.
- I am not sure why it is relevant that CIFAR-100 has more than 42 classes?
- The abstract is a bit unclear on the “predominant” “state-of-the-art” technique, and also does not highlight the data augmentation part in the list of contributions, while the squared loss is listed twice.
- I also cannot match the 52% adv accuracy from the abstract to the right table.

Conclusion:
Overall, I am not convinced by this paper in several regards. First, I find the experiments extremely hard to follow and map which protocol is used in which table. Second, baselines vary a lot and I am actually not sure what would be the right baseline. With these two points, results are hard to judge. Third, I feel hyper-parameters are not discussed, no ablation is given and some details of the proposed “protocol” is also unclear.

---

### Review · Reviewer_1YJd · 2023-08-01

**Summary Of Contributions:**

This paper proposes SimpleAT, a simple but effective adversarial training method that uses a rescaled square loss function, cyclic learning rates, and data augmentation. The authors show square loss is more effective than cross-entropy loss for adversarial training. The authors also demonstrate that the proposed method matches or exceeds state-of-the-art methods on CIFAR and SVHN datasets, and provides analysis showing SimpleAT reduces variance and overconfidence on adversarial examples.


**Audience:**

Yes

**Broader Impact Concerns:**

I don't have any concerns about the ethical implications of the work that would require adding a Broader Impact Statement

**Claims And Evidence:**

Yes

**Requested Changes:**

* I think it's better to provide more explanations and intuitions on why rescaled square loss function is a better choice than cross-entropy loss.
* Minor: Please refrain from only using color to distinguish curves as in Figures 1, 2, 3, 4, 5, 6, and 8, as it is not friendly to readers with color blindness.
* There are some typos or latex mis-ref in the manuscript. For example, "??" appears at the end of section 1. Also, "Ours" and \midrule in Table 2 seem to be in the wrong place. I suggest the authors to do a proofreading.

**Strengths And Weaknesses:**

Strengths:
* This paper provides some key insights around rescaled square loss being better than cross-entropy loss for adversarial training, and how cyclic learning rates and data augmentation help.
* The experimental results show that SimpleAT is comparable to those of the model trained with state-of-the-art techniques.

Weakness
* From the experimental results the proposed method seems to be on par with existing techniques in general. So the practical benefit of using SimpleAT is uncertain to me. Is it because SimpleAT runs faster?

---

### Review · Reviewer_8qSk · 2023-08-03

**Summary Of Contributions:**

This paper proposed a simple adversarial training pipeline to achieve slightly better performance than existing adversarial training pipeline. The core contribution of this work lies in ensembling multiple previous observation in the literature, e.g., competitive performance of square loss, and combining them in one method. For example, the work discovered better performance of rescale square loss over cross-entropy loss and further amplifies it performance with common bag-of-tricks such as data augmentations, weight averaging, and learning rate ablations. The evaluation pipeline is very rigorous as individual design choices in the experiments are carefully ablated and detailed comparison with previous work are provided. To improve interpretability of proposed methods, authors also provide a bias-variance analysis and breakdown of the proposed loss that indicates similarity to a regularized non-softmax loss.

**Audience:**

Yes

**Broader Impact Concerns:**

The paper focuses on fundamental min-max robust learning challenges in deep learning. I don’t see any ethical broader impact concerns with the paper.


**Claims And Evidence:**

Yes

**Requested Changes:**

- Validating generalizability of square loss: While the three datasets considered in the paper are common to use, it is critical to validate the approach on a larger dataset to ensure it’s generalizability. I strongly encourage authors to validate the approach on ImageNet, even if the compute constraints allow single-step adversarial training.
- Loss function beyond square loss: Consider at least another family of loss functions, e.g., hinge loss. Ideally an ablation with interpolation between loss function, e.g., cross-entropy and square loss, through linear interpolation would best highlight the tradeoffs of each loss function.
- In fig.3 and 4, consider provind the plots for loss values in addition to the current 0-1 accuracy plots.

Writing concern/suggestions:
- Page-2: Section ?? concludes this paper
- Page-3: Clearly specify that the attacks formulation is for the linf threat model
- Eq. 8: The rhs part of the equation is incorrect. It should be k^2 and it should be multiplied with both terms in rhs.
- Summarize the final methods in a bounding box at the end paper. Though there are extensive ablations across design choices, the final recommended setup is currently not clear.
- Consider grounding the final results as leaderboard ranking (e.g., in robustbench) to simplify comparison with other works.
- To quantify the rigor in evaluation, consider reporting the gpu hours spent in experiments.


**Strengths And Weaknesses:**

This paper carefully validates the success of proposed work on common vision datasets and provides rigorous ablation study to evaluate the impact of each component. While the work would certainly be insightful for the adversarial training community, I am not convinced that that approach is necessarily better than baseline adversarial training techniques.

*Diminishing returns.* While the gains of proposed approach over baselines are substantial in individual small-scale ablations (mostly on resnet18), there is diminishing return when scaled to state-of-the-art adversarial training pipeline. When combined with extra data and larger networks (wrn-70-16) in table 5, the proposed approach (and even other baseline) don’t outperform the simple cross-entropy loss baseline. While the former(RSL-2-1) achieves slightly better robust accuracy, the latter achieves higher clean accuracy indicating a simple clean-robust accuracy tradeoff. When compared under similar clean accuracy (RSL-5-1), the proposed approach doesn’t outperform baseline CEL in robust accuracy.

*Relationship with hinge loss.* In parallel to cross entropy loss, there are a lot of papers on alternative loss formulation [1, 2]. Currently the work shows success of square loss in adversarial training, but lacks rigorous insights on why this particular loss function succeeds? For example, if square loss is successful in mitigating overfitting, would a similar hinge loss (L1 norm) be also successful?

*Missing comparison with non-overfitting based baselines.* In fig 2.c, it’s evident that both the baseline cross-entropy loss and proposed square loss have similar training dynamics, except near the end, where the cross-entropy loss overfits. Wouldn’t a non-overfitting cross-entropy loss pipeline (e.g., with early stopping or with a separate training setup) be appropriate in this comparison?
Similarly in fig. 4, the comparison of RSL (+IDBH + WA) with baselines such as TRADES is unfair. Both IDBH and WA can be applied (in fact are already being applied [3, 4]) with other adversarial training techniques. RSL performance on-par, not significantly better than other baselines in absence of these techniques.

*Generalizability to large scale datasets.* I am concerned that without further evidence on rigorous testing, it’s not clear whether the approach would generalize beyond the small scale datasets considered in the paper. The current setup indeed considers TinyImageNet (lower resolution and 1/10 size of ImageNet), which has similar complexity as cifar100 dataset. However multiple works have shown that often individual small scale algorithmic innovation may underperform than simple baselines in large scale experiments [5].

1. https://en.wikipedia.org/wiki/Hinge_loss
2. Barron, Jonathan T. "A general and adaptive robust loss function." In Proceedings of the IEEE/CVF Conference on Computer Vision and Pattern Recognition, pp. 4331-4339. 2019.
3. Debenedetti, Edoardo, Vikash Sehwag, and Prateek Mittal. "A light recipe to train robust vision transformers." In 2023 IEEE Conference on Secure and Trustworthy Machine Learning (SaTML), pp. 225-253. IEEE, 2023.
4. Pang, Tianyu, Xiao Yang, Yinpeng Dong, Hang Su, and Jun Zhu. "Bag of tricks for adversarial training." arXiv preprint arXiv:2010.00467 (2020).
5. Kaddour, Jean, Oscar Key, Piotr Nawrot, Pasquale Minervini, and Matt J. Kusner. "No Train No Gain: Revisiting Efficient Training Algorithms For Transformer-based Language Models." arXiv preprint arXiv:2307.06440 (2023).

---

### Note · Authors · 2023-08-29

**Comment:**

Following discussions with all authors,  we would like to formally withdraw the paper titled "Rethinking Adversarial Training with A Simple Baseline". All of the reviewers' comments are suggestive, but due to limited computation resources, we cannot complete all these necessary experiments in a short period of time. We will follow their suggestions and try to revise the paper before submitting it to others. Thanks to AE and reviewers for your hard work.

**Withdrawal Confirmation:**

I have read and agree with the venue's withdrawal policy on behalf of myself and my co-authors.